# Effect of Breed on Hematological and Biochemical Parameters of Apparently Healthy Dogs Infected with Zoonotic Pathogens Endemic to the Mediterranean Basin

**DOI:** 10.3390/ani14111516

**Published:** 2024-05-21

**Authors:** Annalisa Amato, Carmelo Cavallo, Pablo Jesús Marín-García, Giovanni Emmanuele, Mario Tomasello, Cristina Tomasella, Viviana Floridia, Luigi Liotta, Lola Llobat

**Affiliations:** 1Department of Veterinary Sciences, University of Messina, 98168 Messina, Italy; annalisa.amato@studenti.unime.it (A.A.); carmelo.cavallo@studenti.unime.it (C.C.); viviana.floridia@studenti.unime.it (V.F.); 2Departamento Producción y Sanidad Animal, Salud Pública y Ciencia y Tecnología de los Alimentos (PASAPTA), Facultad de Veterinaria, Universidad Cardenal Herrera-CEU, CEU Universities, 46113 Valencia, Spain; pablo.maringarcia@uchceu.es; 3BIOGENE, Veterinary Diagnostic Center, 95127 Catania, Italy; giovanniemmanuele@tiscali.it (G.E.); martomasello@gmail.com (M.T.); cristinatomasella90@gmail.com (C.T.); 4Molecular Mechanisns of Zoonotic Diseases (MMOPS) Research Group, Departamento Producción y Sanidad Animal, Salud Pública y Ciencia y Tecnología de los Alimentos (PASAPTA), Facultad de Veterinaria, Universidad Cardenal Herrera-CEU, CEU Universities, 46113 Valencia, Spain

**Keywords:** *Anaplasma*, canine breed, coinfection, *Erlichia*, *Leishmania*, *Rickettsia*, zoonoses

## Abstract

**Simple Summary:**

Vector-borne infectious zoonotic diseases are a relevant problem, not only in veterinary medicine, but also in human medicine. Dogs are considered the main reservoir of these zoonoses and sentinels for the control of these diseases that, due to the increase in temperatures due to climate change, are becoming more and more frequent. Most of these zoonotic diseases are endemic to the Mediterranean Basin, and knowing their prevalence in different canine breeds and other related factors could contribute to controlling these diseases. This paper analyzes the prevalence of different infections and coinfections of zoonotic disease in four different canine breeds, and the results obtained indicate that the most frequent infections was *Leishmania infantum*, followed by *Ricketsia rickesii*. Changes in the hematological and biochemical values in infected dogs of different breeds could suggest a certain resistance against *L. infantum* infection in the Cirneco dell’Etna canine breed, autochthonous of Sicily.

**Abstract:**

Dogs are considered the main reservoir of several zoonoses endemic to the Mediterranean Basin. In this study, a prevalence of infections and coinfections of canine vector-borne diseases was analyzed in apparently healthy dogs of different canine pure breeds in Sicily (Italy), where these diseases are endemic. The seroprevalence of *Leishmania infantum*, *Ricketsia ricketsii*, *Anaplasma phagocytophilum*, and *Erlichia canis* was assessed, as single and coinfections. Biochemical and hematological parameters were evaluated, and epidemiological factors, including sex, age, and canine breed, were recovered. The most frequent infection was *L. infantum* (45.61%), following *R. ricketsii* (36.84%), both as single, double, or triple coinfections. Coinfections change the biochemical and hematological parameters of the host, and canine breeds are related to the infection frequency and the parameters observed during infections. Changes in the complete blood count (CBC) and biochemical values also differ between canine breeds, with the Cirneco dell’Etna dogs infected with *L. infantum* being the animals presenting the most interesting results in our study. High values of RBC, hemoglobin, hematocrit, mean corpuscular hemoglobin (MCH), the albumin/globulin (A/G) ratio, and albumin and low levels of β-2 globulin and γ-globulin were found only in this canine breed, suggesting some resistance to infection in these dogs. Future studies about the immune response of this canine breed could be interesting to determine their possible resistance to zoonotic pathogens, such as *L. infantum*.

## 1. Introduction

Climate change effects, such as the increase in global temperature or a change in precipitation, have important implications for infectious diseases around the world, mainly in vector-borne diseases [1,2]. Other factors such as urbanization, deforestation, or the abundance of reservoir hosts [3] also influence the vectors of zoonotic diseases, increasing their prevalence from the tropics to lower latitudes [4]. These vector-borne zoonotic diseases are caused by parasites, bacteria, or viruses transmitted by hematophagous vectors, and their prevalence has increased in recent years in all species, including companion animals [5].

Dogs, given their relevance as companion animals and the increase in mobility and worldwide distribution, have contributed to the extension of vectors and canine vector-borne diseases (CVBDs) [6] and, at the same time, could serve as efficient sentinels to analyze the epidemiology impact of zoonoses [7]. Within CVBDs, the major public health concern around the world is represented by leishmaniasis [8], followed by anaplasmosis, ehrlichiosis, and rickettsiosis, endemic to the Mediterranean Basin [9,10]. In this region, these zoonotic diseases are caused by the intracellular protozoan *Leishmania infantum* (order Kinetoplastida, family Trypanosomatidae) and tick-borne bacteria *Anaplasma phagocytophilym*, *Erlichia canis* (order Rickettsiales, family Anaplasmataceae), and *Ricketsia ricketsii* (order Rickettsiales, family Rickettsiaceae). Although *L. infantum* has traditionally been considered to be transmitted by phlebotomine sandflies from the Psychodidae family [11], some studies have reported *L. infantum* detection in ticks in Mediterranean countries, such as Israel [12] or Italy [13,14], so coinfection between the pathogens transmitted by ticks is very likely, especially in endemic areas. Some epidemiological factors could influence both the prevalence of certain infections and the hematological changes observed. Epidemiological factors such as sex, lifestyle, or canine breed are related to the seroprevalence of certain pathogens [15]. The effect of canine breed has been observed in the seroprevalence of different pathogens, such as *L. infantum* [16], *A. phagocytophilum* [17], or *E. canis* [18]. In fact, susceptibility or resistance to different infections seems to be correlated with canine breed, with certain breeds being more susceptible, such as Boxers, Doberman Pinschers, English Cocker Spaniels, or German Shepherds [16,19,20], whereas others seem to be more resistant, such as the Ibizan hound or Beagles [20,21,22], depending on the pathogen. Other canine breeds, such as the Cirneco dell’Etna, phylogenetically close to the Ibizan hound [23] or the St. Bernard, phylogenetically close to Retrievers [24], have never been evaluated for any pathogen. However, studies on coinfections in apparently healthy dogs, their prevalence, their hematological and biochemical parameters, and the epidemiological factors related to change in these parameters are scarce. Knowing the differential values of the hematological and biochemical parameters measured in apparently healthy dogs infected and coinfected with these pathogens is relevant to controlling the prevalence and transmission of zoonoses, not only in dogs as the main silent reservoir, but also in humans.

The objective of this study was to analyze the prevalence of CVBDs, the biochemical and hematological changes associated with them, and the effect of canine breed in apparently healthy dogs in Sicily, in Southern Italy.

## 2. Materials and Methods

### 2.1. Animals and Experimental Design

A total of fifty-seven owned dogs were included in this study between September and November 2023. The pure breeds included were the Cirneco dell’Etna (n = 30), the St. Bernard (n = 8), the English Setter (n = 8), and the English Cocker Spaniel (n = 6). Five animal crossbreeds were also included. Animal experiments were conducted in accordance with the Declaration of Helsinki’s ethical principles and approved by the Animal Experimentation Ethics Committee of the Messina University (code 089/2022, 13 December 2022). Informed consent was obtained from the owners of all the participating dogs. All the animals were bred outdoors in the same geographical area in the province of Catania, eastern Sicily (Italy), near the slopes of the volcano Etna. The inclusion criteria were as follows: adult male and female dogs older than 12 months, living outdoors in kennels (excluding pet dogs), with a complete pedigree, and included in the studbook of each breed (for purebred dogs). The exclusion criteria were the following: a pre-existing disease, with clinical signs, or animals receiving some type of treatment.

Epidemiological data of age, sex, breed, vaccination status, and clinical signs were recovered. Only dogs without clinical signs were included, and they were grouped as young (from one to three years), adults (from four to eight years) and senior (more than eight years) [25]. Ten milliliters of whole blood from apparently healthy dogs [26] was taken by jugular venipuncture and transferred into two tubes, one with EDTA for a complete blood count (CBC) (Sysmex XN-1000, Japanese Sysmex company, Kobe, Japan) and total DNA extraction, and another one without anticoagulant, for the detection of specific parasite antibodies, serum electrophoresis (Capillarys 2, Sebia Dubai SA, Dubai, United Arab Emirates), and a biochemistry profile (BT 3500, Biotecnica Instruments, Rome, Italy).

### 2.2. Parasitic, Biochemical, and Hematological Parameters

Serum samples were tested for *Leishmania* spp., *Anaplasma phagocytophilum*, *Ehrlichia canis*, and *Rickettsia ricketsii*. IgG anti-*Leishmania* spp., anti-*A. pahgocytophilum*, anti-*E. canis,* and anti-*R. ricketsii* antibodies were measured by the indirect immunofluorescence test (IFI), and titers ≥ 1:80 for the *Leishmania* spp. and ≥1:50 for the other pathogens were considered seropositive, following the recommendations of the manufacturer.

The complete blood count (CBC) includes red blood cells’ concentration (RBC), hemoglobin, hematocrit, mean corpuscular volume (MCV), mean corpuscular hemoglobin (MCH), mean corpuscular hemoglobin concentration (MCHC), white blood cells’ count (WBC), neutrophils, lymphocytes, monocytes, eosinophils, and platelets. The biochemistry profile includes glucose, creatinine, urea, total cholesterol, aspartate aminotransferase (AST), alanine aminotransferase (ALT), total protein, albumin/globulin (A/G) ratio, albumin, and alpha-1, alpha-2, beta-1, and beta-2 globulins. The biochemical and hematological parameters were considered altered when they were outside the reference intervals, and the serum protein electrophoretic patterns were defined in accordance with the published guidelines [27].

### 2.3. Blood DNA Extraction and L. infantum Real-Time Polymerase Chain Reaction

To confirm *L. infantum* infection, the DNA was extracted from whole blood using the Invitrogen PureLinkTM Genomic DNA Mini Kit (Thermo Fisher Scientific, Waltham, MA, USA), following the manufacturer’s instructions. The DNA was quantified using a Nanodrop spectrophotometer (Thermo Fisher Scientific, Waltham, MA, USA), and only samples with an A260/A280 ratio > 1.8 were used. RT-PCR was performed by the amplification of kinetoplast minicircle DNA using the primers 5′-GGCGTTCTGCGAAAACCG-3′ and 5′-AAAATGGCATTTTCGGGCC-3′ and the TaqMan probe 5′-FAM-TGGGTGCAGAAATCCCGTTCA-3′. The reaction was carried out in a 20 µL solution containing 0.3 µM of each primer, 0.25 µM of probe, and 20 ng of DNA. The conditions were a first step of UNG at 50 °C for 150 s, a second step of denaturation for 10 min at 95 °C, 40 cycles of denaturation at 95 °C for 15 s, and annealing-polymerization at 60 °C for 35 s [28].

### 2.4. Statistical Analysis

The normality and homoscedasticity of quantitative data were checked by the Shapiro–Wilks and Levene tests, respectively. The quantitative variables were assessed using the *t*-test and the Mann–Whitney U test for normally or non-normally distributed data, respectively, in variables with two categories, using ANOVA (normally distributed) or the Kruskal–Wallis test (non-normally distributed) with more than two categories. The qualitative variables were assessed using Fisher’s exact test or the chi-square test for two nominal variables or more, respectively. A *p*-value < 0.05 was considered statistically significant, and the statistical analysis was performed using the R software and the Rcmdr package.

## 3. Results

Nineteen male dogs (33.33%) and thirty-eight female dogs (66.67%) were included in this study. Regarding the age groups, twenty-seven dogs (47.37%) were young, and twenty-one dogs (36.84%) were adult. All the animals had completed their standard vaccination, and only three dogs (5.26%) were anti-*Leishmania* vaccinated.

From the 57 dogs analyzed, 44 were seropositive for one or more of the pathogens studied (73.68%), 24 of them with a single infection (42.11%), with *L. infantum* being the most common (45.61%, n = 26), followed by *R. ricketsii* (36.84%, n = 21), *A. phagocytophilum* (24.56%, n = 14), and *E. canis* (3.51%, n = 2). Only 9 of the 57 dogs analyzed were positive for *L. infantum* by PCR (15.79%). Double and triple coinfections were found in 31.58% and 3.51% of the dogs, respectively, with *L. infantum*/*R. ricketsii* being the most prevalent. *E. canis* infection was found only in two dogs, with a double- and triple-infection status (Table 1). No statistical association was observed between coinfections.

Sex, age, anti-*Leishmania* vaccination status, and the use of repellent did not influence infection or coinfection. Regarding the pathogens found in the different canine breeds studied, St Bernand dogs were only infected by *L. infantum*, whereas all the pathogens analyzed, except for *E. canis*, were found in the English Setter and English cocker Spaniel specimens. In the crossbreed dogs, *L. infantum* and *R. ricketsii* appeared, whereas the Cirneco dell’Etna presented all the pathogens included in this study (Table 2). Canine breed was correlated to the presence of single and multiple infections, with the Cirneco dell’Etna breed being the most affected by one (15.8%), two (15.8%), or three (3.5%) infections simultaneously.

Our CBC analysis indicated that only the MCHC presented differences between the animals not infected, those infected by one pathogen, or those coinfected by more than one pathogen (Table 3), whereas the biochemical analysis showed that the two aminotransferases (AST and ALT) and the A/G ratio decreased in the dogs with three coinfection pathogens, and the serum electrophoresis data indicated low levels of albumin and high levels of gamma globulin in these animals (Table 4). Related to the epidemiological factors, the female dogs showed high levels of cholesterol, whereas the male dogs had high levels of gamma globulin. Age was correlated with different parameters evaluated in CBC, so senior dogs had higher levels of beta-1 globulin, gamma globulin, and monocytes and lower levels of basophiles, and adult animals showed high levels of WBC and platelets.

Table 5 and Table 6 show the hematological and biochemical parameters in the dogs not infected or doubly coinfected with different pathogens.

Only the number of lymphocytes, such as in the complete blood count (CBC) parameter, differed between groups. *L. infantum*/*R. ricketsii*-coinfected dogs presented lower values of lymphocytes than dogs not infected or coinfected by *R. ricketsii*/*A. phagocytophilum*, and coinfection by *L. infantum*/*A. phagocytophylum* was related to high levels of lymphocytes (Figure 1).

Several changes were observed in the biochemical parameters evaluated according to the coinfections found. Coinfections increased the total protein values, mainly when the coinfection was *L. infantum*/*R. ricketsii*. The albumin/globulin (A/G) ratio and beta-1 globulin decreased in all coinfections, and this effect was mainly observed with *L. infantum* presence. Similar results were observed for the albumin and gamma globulin values, which decreased in all coinfections, with lower albumin levels for *L. infantum*/*R. ricketsii* coinfection and gamma globulin levels for *L. infantum*/*A. phagocytophilum* coinfection (Figure 2).

The individual analysis of the infective pathogens revealed that, out of the epidemiological data, only canine breed was related to different pathogens’ infection. While *A. pahgocytophilum* and *E. canis* infection was not related to canine breed, *R. ricketsii* infection depended on the breed, with a high prevalence in the English Cocker Spaniel and English Setter dogs. Similarly, *L. infantum* infection was higher in the English Cocker Spaniel and St. Bernard dogs (*p* < 0.05). Related to the parameters analyzed, several changes were observed in *L. infantum* infection, both in the CBC (Figure 3) and in the biochemical values (Figure 4). *A. pahgocytophilum* infection increased the glucose and platelets, whereas *R. ricketsii* infection was correlated with an increase in AST transaminases.

Infected dogs of a different age or sex presented the same parameters, whereas the hematological and biochemical data differed according to canine breed and pathogen. The Cirneco dell’Etna dogs infected with *L. infantum* or *R. ricketsii* showed higher values of RBC, hemoglobin, and hematocrit than the dogs of other breeds. *L. infantum* infection in Cirneco dell’Etna dogs also increased the MCV and MCH values and decreased the MCHC according to the data found in the other breeds. On the contrary, the Cirneco dell’Etna dogs infected with *R. ricketsii* or *A. phagocytophilum* showed lower values of WBC, neutrophils, and lymphocytes. This last infection also decreased the MCH in this canine breed (Table 7). This canine breed also presented biochemical parameters different to those of other breeds, mainly in *L. infantum*-infected animals. Their values of urea, AST aminotransferase, the A/G ratio, and albumin were higher than those of other canine breeds, and beta-2 globulin and gamma globulin were lower. *R. ricketsii*-infected dogs of this breed showed higher values of AST aminotransferase, and *A. phagocytophilum*-infected dogs presented higher levels of alpha-2 globulin than other canine breeds. In the St. Bernard dogs, higher levels of creatinine and cholesterol and low levels of alpha-2 globulin were found in *L. infantum* and *R. ricketsii* infections, respectively, compared to other canine breeds. English Cocker Spaniel dogs had lower values of urea and AST aminotransferase in the *L. infantum*-infected animals and lower values of total protein and AST aminotransferase during *R. ricketsii* infection than other canine breeds (Table 8).

## 4. Discussion

The results of the present work showed a high prevalence of single and double infections by zoonotic pathogens in apparently healthy dogs, with a higher prevalence of *L. infantum* infection. Canine breed seemed to influence *L. infantum* and *R. ricketsii* infection, making it so that these two pathogens had a high occurrence in English Cocker Spaniel dogs. Infection with *L. infantum* was also elevated in St. Bernard dogs, whereas the number of English Setter dogs infected with *R. ricketsii* was higher than that of the other breeds. Furthermore, the infective pathogen determined the hematological and biochemical parameters in the apparently healthy dogs, with differences between the canine breeds. The Cirneco dell’Etna dogs were the canine breed with the most different values in terms of the hematological and biochemical parameters in infected dogs, mainly with *L. infantum* infection, and the hematological and biochemical changes depended on the number of infections and the pathogens responsible for the infection in all the animals included. In agreement with other studies carried out in this region, the number of dogs with one or more infections was elevated [30]. The high seroprevalence of *L. infantum* infection found agrees with data observed in other reported studies on Mediterranean areas such as the Balearic Islands [31] or Central Italy [32]. The differences observed between the different *Leishmania* detection methods could be explain due to the fact that anti-*Leishmania* antibodies seemed to increase after the sandfly season, whereas, according to other detection methods, this parameter does not change, probably because the humoral immune response is more sensible to re-exposure [33]. The seroprevalence of *R. ricketsii* and *A. phagocytophilum* infection was similar to data observed in private kennels in this region [30]. *E. canis* infection was lower and always appeared in double or triple coinfections. Similar results have been found in Iran [34], whereas other authors have reported a higher prevalence of *E. canis* and a lower prevalence of *A. phagocytophilum* [35,36]. Serological cross-reaction exists in these two species but not in the other pathogens studied [37], so these results could be the result of this cross-reaction.

The most common double coinfection was *L. infantum*/*R. ricketsii* (17.54%). Although phlebotomine sandflies have been considered the main vector of the *Leishmania* spp. in dogs and humans [38], different studies have found the DNA of *L. infantum* in *Ripichephalus sanguineus* [39,40] and *Ixodes ricinus* ticks [41,42] in the Mediterranean Basin. Given that these two tick species are related to *R. ricketsii* infection in dogs and humans [43,44], the high number of coinfections found with these two parasites could be due to the simultaneous transmission of both through the same vector. Like sandflies, the number of ticks and their human affinity increase with warmer temperatures [45]; therefore, climate change and the associated increase in temperatures will most likely increase the cases of these zoonoses in humans.

The number of infections changed the hematological and biochemical parameters, so MCHC and transaminases increased in the animals presenting more than one pathogen, and gamma globulin increased when three pathogens were simultaneously present. However, since the number of animals with three simultaneous infections was low (n = 2), we must take this result with caution and carry out more studies to corroborate it. The effect of the hematological parameters in coinfections has been observed previously in dogs infected with *A. platys* and/or *E. canis*, increasing anemia and thrombocytopenia [46]. Thrombocytopenia is common in single *Anaplasma* spp. infections or in coinfections with other pathogens [47], and other hematological abnormalities related to anemia and the increase in transaminases are common in coinfected dogs with clinical signs [48]. Low A/G ratios and hypergammaglobulinemia are typically found in *Leishmania* spp. infections with clinical signs [49], and this abnormality increases with coinfections [50], whereas hypergammaglobulinemia has not been observed in apparently healthy animals infected only by *A. phagocytophilum* [51]. Although there are few studies on the hematological and biochemical values in apparently healthy dogs coinfected with the pathogens studied, our results seem to indicate that coinfections increase the abnormalities in some of these parameters, which could accelerate the appearance of clinical signs.

While neither age nor sex were relevant factors for the appearance of infections and/or coinfections, the dog breed was a determining factor for *L. infantum* and *R. ricketsii* infection, being most prevalent in the English Cocker Spaniel dogs. Differences in the seroprevalence of *L. infantum* infection between canine breeds have been reported previously, with a high prevalence in Boxers, Dobermann Pinschers [16], and English Cocker Spaniels [52], according to our results. In this paper, we also found a high prevalence, for the first time, in the St. Bernard breed, classified in group two of the Federation Cynologique Internationale (FCI) [53], together with Boxers and Dobermann Pinschers, canine breeds with a high seroprevalence of *L. infantum* infection [16]. Related to *R. ricketsii* infection, no data about a different prevalence in canine breeds exist, but several studies have reported a differential prevalence according to lifestyle (outdoor more than indoor) [54,55] and crossbreeds compared to pure breeds (more prevalent in the former) [56].

*L. infantum* infection decreased the hematocrit, hemoglobin, A/G ratio, albumin, total protein, and ALT transaminase and increased alpha-2, beta-1, beta-2, and gamma globulins in apparently healthy dogs. These results are in agreement with [57] who reported the same changes between seropositive healthy and sick dogs. The above-mentioned authors did not find differences in the eosinophil and MCH values, whereas our results showed an increase in these parameters in the infected dogs. A review published by [58] explained how increases in eosinophils’ function and number are related to a high protection against different pathogens, including the *Leishmania* spp., which could explain the high values of eosinophils in infected dogs.

Previous studies have related coinfections with *A. platys* and *E. canis* and double or triple coinfections to a decrease in platelets [46,59]. Our results showed that *A. phagocytophilum* infection increased glucose and platelets, whereas *R. ricketsii* infection was correlated with an increase in transaminases. These results could explain why *A. platys* infects platelets while the target cells of *A. phagocytophilum* are neutrophils [60]. Increase in AST transaminase related to parasite infection have been observed previously in cattle infected with the *Anaplasma* spp. [61], probably as the first sign of poor liver function.

The analysis of pathogen infection and the epidemiological data showed that sex and age did not change the biochemical and hematological parameters, which were the same for female and male dogs and for young, adult, and elder animals. However, the Cirneco dell’Etna dogs presented different values during *L. infantum*, *R. ricketsii*, and *A. phagocytophilum* infection. The most relevant changes were related to *L. infantum* infection, during which these animals had high values of RBC, hemoglobin, hematocrit, MCH, A/G ratio, and albumin and low values of beta-2 globulin and gamma globulin. Previous studies carried out in other greyhound breeds reported that greyhounds have higher hematocrit and hemoglobin concentrations [62] as well as higher MCH levels and MCHCs [63]. Although these studies were conducted in healthy dogs, these results could have indicated that greyhounds have different hematological and/or biochemical parameters compared to other canine breeds, so further studies with a high number of infected and non-infected Cirneco dell’Etna dogs are necessary to elucidate whether these differences are due to the canine breed or infection by different parasites. Our results found that these parameters differ significantly from the results published based on healthy and sick infected dogs [57], which could indicate resistance to this infection in this breed or a greater ability to control the clinical signs. Less arresting were the differences found in this dog breed in infections with the other pathogens studied, even if the *R. ricketsii*-infected dogs had surprisingly high levels of RBC, hemoglobin, and hematocrit, which could have indicated a certain ability of this dog breed to control intracellular infections. The mechanisms to control *Leishmania* spp. and *Rickettsia* spp. infections are similar and involve several cytokines related to the Th1 immune response, such as interferon gamma (IFN-γ) [64,65], tumor necrosis factor-alpha [66,67], and other interleukins. These cytokines, mainly IFN-γ, play a relevant role in macrophage activation via nitric oxide, as in vitro and in vivo studies have demonstrated [68]. The cell membrane receptor of IFN-γ in macrophages stimulates the JAK-STA-1 pathway [69], and the activation of macrophages via IFN-γ regulates positively MHC II, increases ROS and NOS production, and induces autophagy for intracellular pathogens’ elimination [70]. In fact, high serum levels of IFN-γ and haplotypes in genes encoding this cytokine have been related to resistance to *L. infantum* infection in Ibizan hound dogs [71,72]. No studies have been conducted in Cirneco dell’Etna dogs related to cytokines serum levels, immune response, and resistance to intracellular infection. In view of the results found in this study, and given that this breed has genomic similarities with the Ibizan hound [23], a more exhaustive analysis of this dog breed regarding its immune response to infections would be interesting.

## 5. Conclusions

*L. infantum* and *R. ricketsii* infections were the most prevalent infections in dogs in Sicily (Italy), both as single infections and as coinfections together or with another pathogen. The occurrence of the pathogens studied, as well as the abnormal biochemical and hematological parameters, differed between canine breeds. These values changed with single, double, or triple coinfections, indicating that, in apparently healthy dogs, certain changes had already been detected if there were coinfections, probably because the presence of another pathogen or pathogens hindered the host’s immune capacity to control the primary infection. Changes in CBC and biochemical values also differed between canine breeds, with the Cirneco dell’Etna dogs infected with *L. infantum* being the animals with the most interesting results. High values of RBC, hemoglobin, hematocrit, MCH, A/G ratio, and albumin and low levels of beta-2 globulin and gamma globulin were found in the Cirneco dell’Etna dogs infected with *L. infantum*, and these results were not found in the other canine breeds. These results could indicate a great ability to control the clinical signs related to infection and a better immune system response in this canine breed. A possible limitation of this study is the limited number of animals included in it, so it would be interesting to carry out new studies with a larger number of individuals, mainly from the native Cirneco dell’Etna breed. Inasmuch as this breed is genetically related to the Ibizan hound, traditionally considered resistant to *Leishmania* spp. infection, studies on the immune response of this native breed of the Sicilian region would be interesting to evaluate its ability to control infection.

## Figures and Tables

**Figure 1 animals-14-01516-f001:**
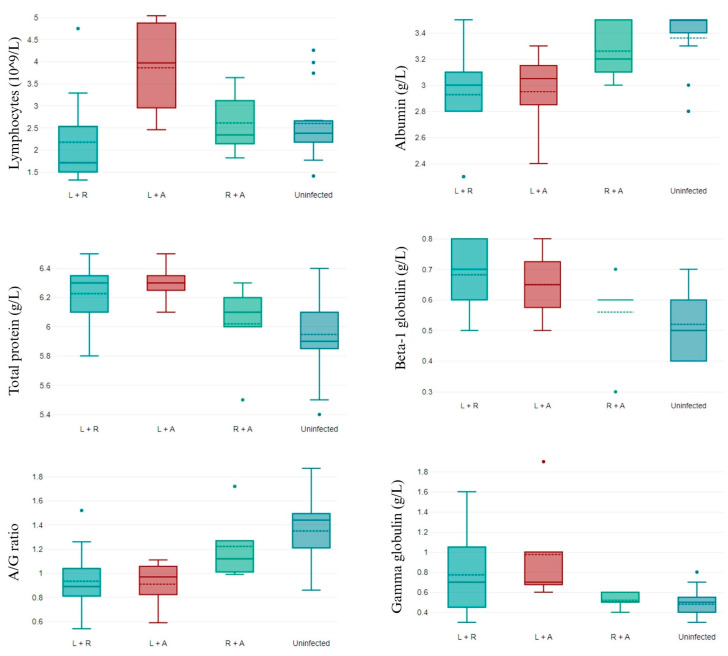
CBC parameters that differ in non-infected and double-coinfected dogs. L + R: *L. infantum* and *R. ricketsii* coinfection; L + A: *L. infantum* and *A. phagocytophilum* coinfection; R + A: *R. ricketsii* and *A. phagocytophilum* coinfection; and A/G ratio: albumin/globulin ratio. Data shown as mean ± SD.

**Figure 2 animals-14-01516-f002:**
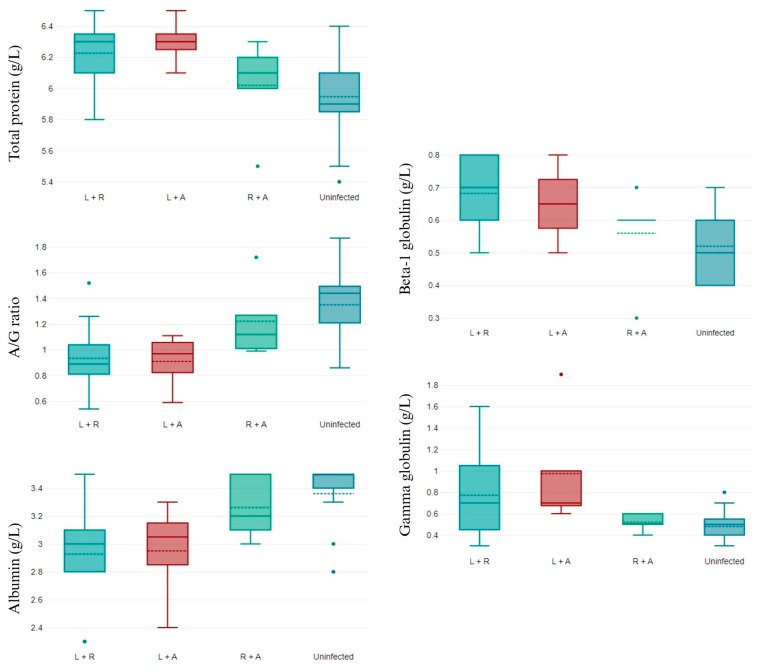
Biochemical parameters that differ in non-infected and double-coinfected dogs. L + R: *L. infantum* and *R. ricketsii* coinfection; L + A: *L. infantum* and *A. phagocytophilum* coinfection; R + A: *R. ricketsii* and *A. phagocytophilum* coinfection; and A/G ratio: albumin/globulin ratio. Data shown as mean ± SD.

**Figure 3 animals-14-01516-f003:**
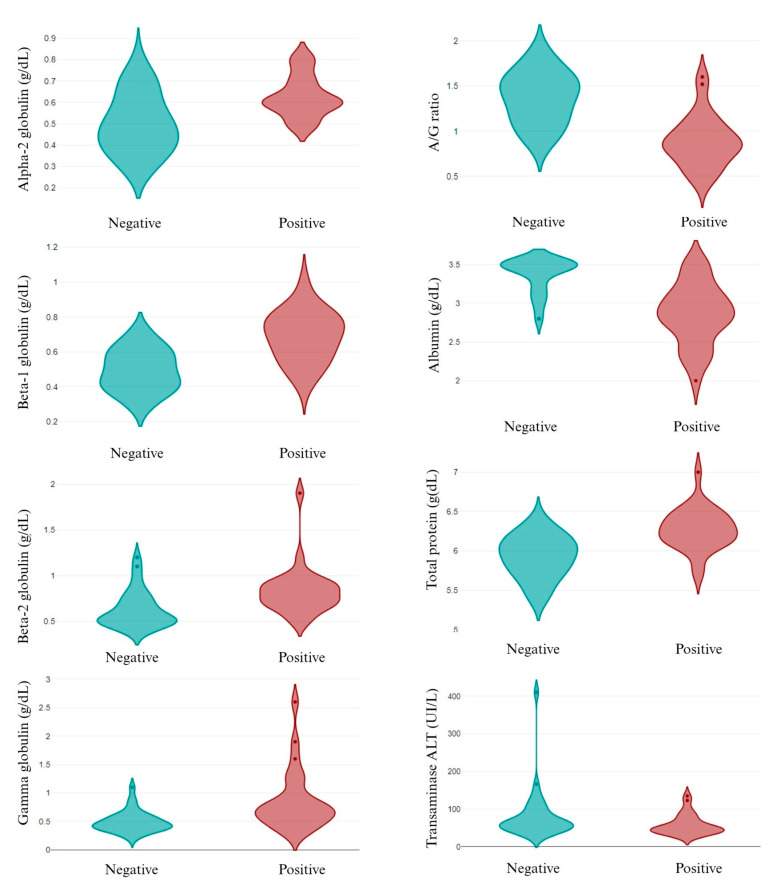
CBC in *L. infantum* non-infected (negative) and infected (positive) dogs. A/G ratio: albumin/globulin ratio.

**Figure 4 animals-14-01516-f004:**
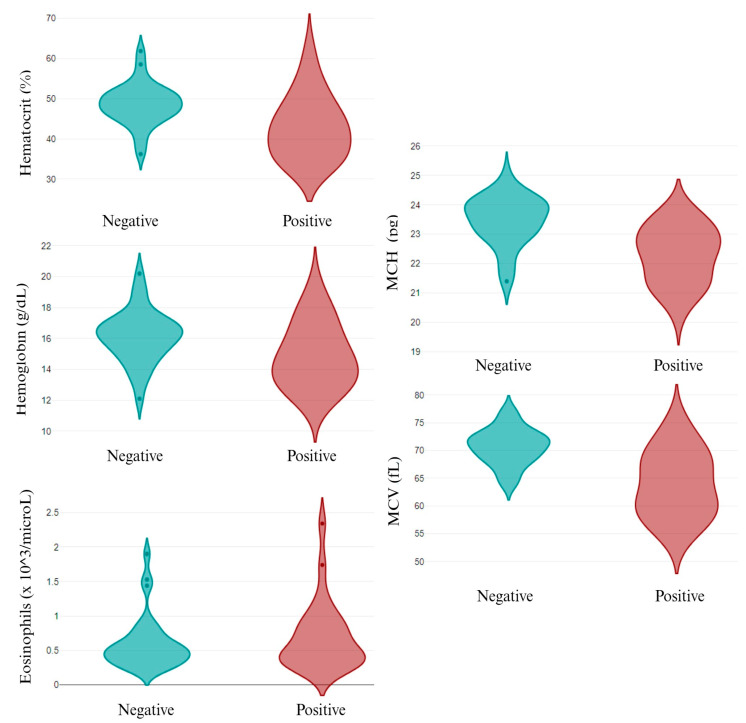
Differential biochemical parameters in *L. infantum* non-infected (negative) and infected (positive) dogs. MCH: mean corpuscular hemoglobin; and MCV: mean corpuscular volume.

**Table 1 animals-14-01516-t001:** Occurrence of infections and coinfections in the dogs studied.

Type of Infection	Pathogen Species Found	Number of Positive Dogs (%)
Single infection	*L. infantum*	11 (19.30%)
	*R. ricketsii*	7 (12.28%)
	*A. phagocytophilum*	6 (10.53%)
	*E. canis*	0
	Total single infection	24 (42.11%)
Double infection	*L. infantum* + *R. ricketsii*	11 (17.54%)
	*L. infantum* + *A.phagocytophilum*	3 (5.26%)
	*R. ricketsii* + *A.phagocytophilum*	3 (5.26%)
	*A. phagocytophilum* + *E. canis*	1 (1.75)
	Total double infection	18 (31.58%)
Triple infection	*L. infantum* + *R. ricketsii* + *A.phagocytophilum*	1 (1.75%)
	*L. infantum* + *A.phagocytophilum* + *E. canis*	1 (1.75%)
	Total triple infection	2 (3.51%)

**Table 2 animals-14-01516-t002:** Infections found in the different canine breeds included.

Canine Breed	*L. infantum* Infection (Number of Dogs/Total Dogs)	*R. ricketsii* Infection (Number of Dogs/Total Dogs)	*A. pahgocytophylum* Infection (Number of Dogs/Total Dogs)	*E. canis* Infection (Number of Dogs/Total Dogs)
Cirneco dell’Etna	11/30	10/30	11/30	2/30
St. Bernard	7/8	0/8	0/8	0/8
English Setter	1/8	7/8	2/8	0/8
English Cocker Spaniel	6/6	3/6	1/6	0/6
Crossbred	1/5	1/5	0/5	0/5

**Table 3 animals-14-01516-t003:** Complete blood count (CBC) parameters of infected and coinfected dogs. Data shown as mean ± SD. A different superscript indicates significant differences (*p* < 0.05).

CBC Parameters *	Intervals [29]	Not Infected	One Infection	Two Infections	Three Infections	*p*-Value
RBC (10^6^/µL)	5.0–8.1	7.03 ± 0.83	6.58 ± 0.67	6.77 ± 0.79	7.06 ± 0.54	0.315
Hemoglobin (g/dL)	12.0–18.0	16.31 ± 2.09	15.12 ± 1.72	15.44 ± 2.11	15.60 ± 1.98	0.325
Hematocrit (%)	37.0–55.0	49.24 ± 6.10	43.71 ± 6.76	45.63 ± 7.85	47.55 ± 4.88	0.121
MCV (fL)	60.0–75.0	69.9 ± 3.96	66.27 ± 6.80	67.08 ± 5.65	67.25 ± 1.77	0.412
MCH (pg)	19.0–25.0	23.15 ± 0.93	22.96 ± 1.24	22.76 ± 0.94	22.05 ± 1.06	0.476
MCHC (g/dL)	32.0–36.0	33.14 ± 1.62 ^a^	34.88 ± 2.17 ^b^	34.06 ± 1.91 ^ab^	32.75 ± 0.78 ^ab^	**<0.05**
WBC (10^9^/L)	6.0–15.0	10.14 ± 3.70	10.02 ± 2.43	10.55 ± 3.25	7.57 ± 1.88	0.529
Neutrophils (10^9^/L)	3.6–8.0	6.64 ± 3.40	6.07 ± 1.78	6.80 ± 2.02	3.81 ± 0.42	0.133
Lymphocytes(10^9^/L)	0.8–4.1	2.58 ± 0.78	2.91 ± 0.95	2.80 ± 1.37	3.16 ± 2.35	0.708
Monocytes (10^9^/L)	0.1–0.6	0.29 ± 0.19	0.67 ± 1.71	0.36 ± 0.25	0.22 ± 0.16	0.781
Eosinophils (10^9^/L)	0.1–1.0	0.60 ± 0.46	0.72 ± 0.51	0.62 ± 0.35	0.37 ± 0.10	0.532
Platelets (10^3^/µL)	150.0–500.0	236.25 ± 83.49	270.14 ± 131.51	260.12 ± 127.23	319.00 ± 197.99	0.861

* RBC: red blood cells’ concentration; MCV: mean corpuscular volume; MCH: mean corpuscular hemoglobin; MCHC: mean corpuscular hemoglobin concentration; and WBC: white cells’ blood count.

**Table 4 animals-14-01516-t004:** Biochemical parameters of the infected and coinfected dogs. Data shown as mean ± SD. A different superscript in the same row indicates significant differences (*p* < 0.05).

Parameters		Intervals [29]	Not Infected	One Infection	Two Infections	Three Infections	*p*-Value
Glucose (mg/dL)		60.0–120.0	72.25 ± 8.14	70.95 ± 6.16	73.35 ± 11.43	70.00 ± 2.82	0.982
Creatinine (mg/dL)		<2.0	1.01 ± 0.15	0.97 ± 0.18	0.94 ± 0.18	0.80 ± 0.14	0.319
Urea (mg/dL)		10.0–25.0	13.00 ± 2.78	12.95 ± 2.93	13.18 ± 3.91	14.50 ± 4.95	0.905
Cholesterol (mg/dL)		120.0–330.0	217.50 ± 61.83	227.41 ± 60.65	208.24 ± 53.34	171.50 ± 62.93	0.526
Transaminases AST (UI/L)		<90.0	40.43 ± 19.50 ^a^	49.91 ± 20.47 ^a^	62.35 ± 29.33 ^b^	50.00 ± 18.38 ^b^	**<0.05**
Transaminases ALT (UI/L)		<100.0	94.50 ± 90.72 ^a^	57.36 ± 28.08 ^a^	70.52 ± 28.91 ^a^	35.00 ± 4.25 ^b^	**<0.05**
Total protein (g/L)		5.5–7.8	5.92 ± 0.28	6.11 ± 0.36	6.17 ± 0.22	6.30 ± 0.28	0.072
A/G ratio		>0.9	1.37 ± 0.30 ^a^	1.09 ± 0.40 ^a^	1.06 ± 0.32 ^a^	0.78 ± 0.27 ^b^	**<0.05**
Serum electrophoresis (g/L)	Albumin	2.5–3.5	3.36 ± 0.26 ^a^	3.04 ± 0.43 ^b^	3.09 ± 0.40 ^ab^	2.71 ± 0.42 ^ab^	**<0.05**
	Alpha-1 globulin	0.2–0.5	0.40 ± 0.05	0.40 ± 0.07	0.42 ± 0.08	0.35 ± 0.07	0.501
	Alpha-2 globulin	0.3–1.0	0.53 ± 0.14	0.56 ± 0.16	0.58 ± 0.11	0.60 ± 0.00	0.665
	Beta-1 globulin	0.4–0.9	0.51 ± 0.12	0.59 ± 0.19	0.64 ± 0.15	0.60 ± 0.00	0.121
	Beta-2 globulin	0.5–1.0	0.62 ± 0.23	0.79 ± 0.31	0.74 ± 0.19	0.65 ± 0.07	0.218
	Gamma globulin	0.5–1.2	0.48 ± 0.14 ^a^	0.70 ± 0.49 ^ab^	0.69 ± 0.38 ^ab^	1.35 ± 0.78 ^b^	**<0.05**

**Table 5 animals-14-01516-t005:** Complete blood count (CBC) parameters of non-infected and double-coinfected dogs (*L. infantum* and *A. phagocytophilum* coinfection, *L. infantum* and *R. ricketsii* coinfection, and *R. ricketsii* and *A. phagocytophilum* coinfection). Data shown as mean ± SD. A different superscript indicates significant differences (*p* < 0.05).

CBC Parameters *	Intervals [29]	Not Infected	L + A	L + R	R + A	*p*-Value
RBC (10^6^/µL)	5.0–8.1	7.03 ± 0.83	6.74 ± 0.61	6.91 ± 0.88	6.97 ± 0.62	0.958
Hemoglobin (g/dL)	12.0–18.0	16.31 ± 2.09	15.23 ± 1.90	15.51 ± 2.45	16.22 ± 1.14	0.741
Hematocrit (%)	37.0–55.0	49.24 ± 6.10	44.78 ± 6.24	46.14 ± 9.28	48.26 ± 2.81	0.615
MCV (fL)	60.0–75.0	69.9 ± 3.96	66.28 ± 4.56	66.28 ± 6.25	69.36 ± 2.91	0.150
MCH (pg)	19.0–25.0	23.15 ± 0.93	22.58 ± 1.00	22.36 ± 0.87	23.30 ± 0.72	0.08
MCHC (g/dL)	32.0–36.0	33.14 ± 1.62a	34.13 ± 2.24	33.92 ± 2.15	33.58 ± 1.11	0.609
WBC (10^9^/L)	6.0–15.0	10.14 ± 3.70	10.30 ± 1.66	9.32 ± 3.25	8.92 ± 1.35	0.798
Neutrophils (10^9^/L)	3.6–8.0	6.64 ± 3.40	5.58 ± 1.57	6.21 ± 2.01	5.35 ± 0.58	0.726
Lymphocytes (10^9^/L)	0.8–4.1	2.58 ± 0.78 ^ab^	3.86 ± 1.27 ^a^	2.17 ± 1.06 ^b^	2.61 ± 0.75 ^ab^	**<0.05**
Monocytes (10^9^/L)	0.1–0.6	0.29 ± 0.19	0.43 ± 0.44	0.30 ± 0.16	0.32 ± 0.13	0.720
Eosinophils (10^9^/L)	0.1–1.0	0.60 ± 0.46	0.63 ± 0.22	0.61 ± 0.42	0.62 ± 0.52	0.998
Platelets (10^3^/µL)	150.0−500.0	236.25 ± 83.49	295.50 ± 97.26	253.91 ± 158.86	346.00 ± 65.66	0.275

* RBC: red blood cells’ concentration; MCV: mean corpuscular volume; MCH: mean corpuscular hemoglobin; MCHC: mean corpuscular hemoglobin concentration; WBC: white blood cells’ count; L: *L. infantum* infection; A: *A. phagocytophilum* infection; and R: *R. ricketsii* infection.

**Table 6 animals-14-01516-t006:** Biochemical parameters of non-infected and double-coinfected dogs (*L. infantum* and *A. phagocytophilum* coinfection, *L. infantum* and *R. ricketsii* coinfection, and *R. ricketsii* and *A. phagocytophilum* coinfection). Data shown as mean ± SD. A different superscript in a row indicates significant differences (*p* < 0.05).

Parameters *		Intervals [29]	Not Infected	L + A	L + R	R + A	*p*-Value
Glucose (mg/dL)		60.0–120.0	72.25 ± 8.14	74.75 ± 11.87	69.73 ± 8.34	76.60 ± 5.18	0.451
Creatinine (mg/dL)		<2.0	1.01 ± 0.15	1.00 ± 0.18	0.88 ± 0.19	0.90 ± 0.07	0.198
Urea (mg/dL)		10.0–25.0	13.00 ± 2.78	15.00 ± 5.94	12.82 ± 3.71	12.60 ± 3.13	0.733
Cholesterol (mg/dL)		120.0–330.0	217.50 ± 61.83	156.50 ± 21.42	211.36 ± 40.45	190.60 ± 90.27	0.187
Transaminases AST (UI/L)		<90.0	40.43 ± 19.50	59.75 ± 27.44	63.82 ± 31.19	42.60 ± 11.13	0.363
Transaminases ALT (UI/L)		<100.0	94.50 ± 90.72	59.25 ± 22.16	73.55 ± 33.76	57.40 ± 34.63	0.560
Total protein (g/L)		5.5–7.8	5.92 ± 0.28 ^a^	6.30 ± 0.16 ^ab^	6.23 ± 0.21 ^b^	6.02 ± 0.31 ^ab^	**<0.05**
A/G ratio		>0.9	1.37 ± 0.30 ^a^	0.91 ± 0.23 ^a^	0.93 ± 0.29 ^a^	1.22 ± 0.30 ^ab^	**<0.05**
Serum electrophoresis (g/L)	Albumin	2.5–3.5	3.36 ± 0.26 ^a^	2.95 ± 0.39 ^ab^	2.93 ± 0.40 ^b^	3.26 ± 0.23 ^ab^	**<0.05**
	Alpha-1 globulin	0.2–0.5	0.40 ± 0.05	0.38 ± 0.05	0.44 ± 0.09	0.38 ± 0.04	0.228
	Alpha-2 globulin	0.3–1.0	0.53 ± 0.14	0.63 ± 0.05	0.61 ± 0.07 ^b^	0.56 ± 0.18 ^ab^	0.361
	Beta-1 globulin	0.4–0.9	0.51 ± 0.12 ^a^	0.65 ± 0.13 ^b^	0.68 ± 0.12 ^b^	0.56 ± 0.15 ^ab^	**<0.05**
	Beta-2 globulin	0.5–1.0	0.62 ± 0.23	0.73 ± 0.05	0.79 ± 0.21	0.72 ± 0.15	0.254
	Gamma globulin	0.5–1.2	0.48 ± 0.14 ^a^	0.98 ± 0.62 ^b^	0.77 ± 0.44 ^ab^	0.52 ± 0.08 ^ab^	**<0.05**

* Transaminases AST: aspartate aminotransferase; transaminases ALT: alanine aminotransferase; A/G: albumin/globulin; L: *L. infantum* infection; A: *A. phagocytophilum* infection; and R: *R. ricketsii* infection.

**Table 7 animals-14-01516-t007:** Complete blood count (CBC) parameters in infected dogs according to canine breed. Data shown as mean ± SD. A different superscript in the same row indicates significant differences (*p* < 0.05).

CBC Parameters *	Intervals [29]	*L. infantum* Infection				*R. ricketsii* Infection				*A. phagocytophilum* Infection		
		Cirneco dell’Etna	English Cocker Spaniel	St. Bernard	*p*-Value	Cirneco dell’Etna	English Cocker Spaniel	English Setter	*p*-Value	Cirneco dell’Etna	English Setter	*p*-Value
RBC (10^6^/µL)	5.0–8.1	7.20 ± 0.71 ^a^	6.14 ± 0.48 ^b^	6.41 ± 0.85 ^ab^	**<0.05**	7.40 ± 0.68 ^a^	6.14 ± 0.36 ^b^	6.34 ± 0.29 ^b^	**<0.05**	6.99 ± 0.52	6.26 ± 0.58	0.09
Hemoglobin (g/dL)	12.0–18.0	16.53 ± 1.81 ^a^	13.30 ± 0.88 ^b^	13.96 ± 1.83 ^b^	**<0.05**	17.21 ± 1.57 ^a^	13.07 ± 0.75 ^b^	15.09 ± 0.78 ^b^	**<0.05**	16.15 ± 1.21	15.00 ± 1.41	0.252
Hematocrit (%)	37.0–55.0	50.02 ± 6.05 ^a^	36.47 ± 2.79 ^b^	37.30 ± 4.38 ^b^	**<0.05**	52.60 ± 5.67 ^a^	35.80 ± 2.46 ^b^	44.99 ± 2.28 ^c^	**<0.05**	48.18 ± 2.73	44.45 ± 4.45	0.126
MCV (fL)	60.0–75.0	69.43 ± 3.44 ^a^	59.38 ± 1.54 ^b^	58.36 ± 3.16 ^b^	**<0.05**	71.08 ± 4.17 ^a^	58.23 ± 0.96 ^b^	70.94 ± 1.56 ^a^	**<0.05**	69.00 ± 2.39	71.00 ± 0.57	0.279
MCH (pg)	19.0–25.0	22.95 ± 0.73 ^a^	21.68 ± 0.61 ^b^	21.80 ± 1.09 ^b^	**<0.05**	23.26 ± 0.74 ^a^	21.27 ± 0.12 ^b^	23.79 ± 0.73 ^a^	**<0.05**	23.10 ± 0.79	23.95 ± 0.07	**<0.05**
MCHC (g/dL)	32.0–36.0	33.10 ± 1.51 ^a^	36.50 ± 0.50 ^b^	37.37 ± 1.46 ^b^	**<0.05**	32.89 ± 1.50 ^a^	36.50 ± 0.44 ^b^	33.53 ± 0.34 ^a^	**<0.05**	33.46 ± 0.97	33.75 ± 0.21	0.696
WBC (10^9^/L)	6.0–15.0	9.02 ± 1.82	10.24 ± 3.41	9.99 ± 3.05	0.605	8.32 ± 1.89 ^a^	9.02 ± 4.76 ^ab^	12.98 ± 2.65 ^b^	**<0.05**	9.01 ± 1.40	15.64 ± 2.55	**<0.05**
Neutrophils (10^9^/L)	3.6–8.0	5.49 ± 1.34	6.51 ± 2.16	6.10 ± 2.41	0.654	5.50 ± 1.35 ^a^	5.90 ± 2.90 ^ab^	8.04 ± 1.82 ^b^	**<0.05**	5.36 ± 1.07	10.01 ± 1.80	**<0.05**
Lymphocytes (10^9^/L)	0.8–4.1	2.57 ± 1.13	3.21 ± 1.57	2.59 ± 0.66	0.792	1.97 ± 0.65 ^a^	2.54 ± 1.92 ^ab^	4.02 ± 1.23 ^b^	**<0.05**	2.70 ± 0.94	4.90 ± 1.15	**<0.05**
Monocytes (10^9^/L)	0.1–0.6	0.35 ± 0.27	0.15 ± 0.09	0.41 ± 0.19	0.054	0.31 ± 0.10	0.20 ± 0.11	0.24 ± 0.21	0.472	0.43 ± 0.25	0.14 ± 0.04	0.135
Eosinophils (10^9^/L)	0.1–1.0	0.68 ± 0.32	0.34 ± 0.11	0.86 ± 0.71	0.061	0.53 ± 0.19	0.36 ± 0.06	0.64 ± 0.20	0.134	0.59 ± 0.36	0.55 ± 0.37	0.843
Platelets (10^3^/µL)	150.0–500.0	319.63 ± 205.62	228.00 ± 76.23	212.71 ± 59.88	0.528	235.50 ± 180.76	218.67 ± 69.92	273.29 ± 47.26	0.798	337.18 ± 89.61	247.50 ± 0.71	0.199

* RBC: red blood cells’ concentration; MCV: mean corpuscular volume; MCH: mean corpuscular hemoglobin; MCHC: mean corpuscular hemoglobin concentration; and WBC: white blood cells’ count.

**Table 8 animals-14-01516-t008:** Biochemical parameters in infected dogs according to canine breed. Data shown as mean ± SD. A different superscript in the same row indicates significant differences (*p* < 0.05).

Parameters * (Units)	Intervals [29]		*L. infantum* Infection				*R. ricketsii* Infection				*A. phagocytophilum* Infection		
			Cirneco dell’Etna	English Cocker Spaniel	St. Bernard	*p*-Value	Cirneco dell’Etna	English Cocker Spaniel	English Setter	*p*-Value	Cirneco dell’Etna	English Setter	*p*-Value
Glucose (mg/dL)	60.0–120.0		70.09 ± 9.331	78.00 ± 8.29	68.71 ± 2.06	0.051	69.50 ± 12.83	76.00 ± 6.56	69.29 ± 5.22	0.152	76.09 ± 9.65	71.00 ± 9.90	0.508
Creatinine (mg/dL)	<2.0		0.90 ± 0.16 ^a^	0.75 ± 0.08 ^a^	1.16 ± 0.11 ^b^	**<0.05**	0.90 ± 0.16	0.77 ± 0.12	0.99 ± 0.12	0.08	0.92 ± 0.14	1.10 ± 0.14	0.120
Urea (mg/dL)	10.0–25.0		14.18 ± 4.40 ^a^	10.17 ± 0.41 ^b^	13.14 ± 3.24 ^ab^	**<0.05**	14.30 ± 3.77	10.00 ± 0.00	13.71 ± 2.14	0.06	14.18 ± 4.19	13.50 ± 0.71	0.829
Cholesterol (mg/dL)	120.0–330.0		196.73 ± 50.23 ^a^	185.17 ± 41.77 ^a^	255.86 ± 25.86 ^b^	**<0.05**	233.70 ± 48.97	186.67 ± 20.60	233.57 ± 71.05	0.422	193.27 ± 76.95	211.50 ± 85.56	0.844
Transaminases AST(UI/L)	<90.0		65.18 ± 30.61 ^a^	33.83 ± 5.08 ^b^	49.00 ± 7.16 ^c^	**<0.05**	73.90 ± 32.91 ^a^	35.00 ± 5.29 ^b^	57.57 ± 15.28 ^ab^	**<0.05**	49.27 ± 20.25	76.50 ± 13.44	0.100
Transaminases ALT (UI/L)	<100.0		62.36 ± 28.91	47.33 ± 4.03	44.57 ± 10.21	0.619	77.50 ± 34.89	49.33± 1.53	70.71 ± 15.76	0.209	53.45 ± 26.92	73.00 ± 16.97	0.352
Total protein (g/L)	5.5–7.8		6.15 ± 0.24	6.45 ± 0.31	6.34 ± 0.18	0.053	77.50 ± 34.89 ^ab^	49.33 ± 1.53 ^a^	70.71 ± 15.76 ^b^	**<0.05**	6.09 ± 0.26	6.00 ± 0.42	0.679
A/G ratio	>0.9		1.06 ± 0.30 ^a^	0.71 ± 0.26 ^b^	0.78 ± 0.18 ^b^	**<0.05**	1.24 ± 0.27 ^a^	0.63 ± 0.15 ^b^	1.39 ± 0.38 ^a^	**<0.05**	1.13 ± 0.31	1.40 ± 0.32	0.295
Serum electrophoresis (g/L)	Albumin	2.5–3.5	3.09 ± 0.33 ^a^	2.58 ± 0.47 ^b^	2.74 ± 0.30 ^b^	**<0.05**	3.29 ± 0.23 ^a^	2.47 ± 0.29 ^b^	3.33 ± 0.28 ^a^	**<0.05**	3.16 ± 0.34	3.45 ± 0.07	0.307
	Alpha-1 globulin	0.2–0.5	0.42 ± 0.08	0.38 ± 0.08	0.43 ± 0.05	0.418	0.40 ± 0.08	0.40 ± 0.10	0.43 ± 0.08	0.749	0.38 ± 0.04	0.40 ± 0.00	0.529
	Alpha-2 globulin	0.3–1.0	0.63 ± 0.09	0.62 ± 0.04	0.66 ± 0.14	0.909	0.57 ± 0.11 ^a^	0.60 ± 0.00 ^a^	0.39 ± 0.07 ^b^	**<0.05**	0.57 ± 0.12	0.35 ± 0.07	<0.05
	Beta-1 globulin	0.4–0.9	0.67 ± 0.14	0.67 ± 0.10	0.76 ± 0.17	0.419	0.59 ± 0.17 ^ab^	0.73 ± 0.06 ^a^	0.43 ± 0.08 ^b^	**<0.05**	0.59 ± 0.14	0.45 ± 0.07	0.195
	Beta-2 globulin	0.5–1.0	0.69 ± 0.12 ^a^	0.92 ± 0.17 ^b^	1.01 ±0.41 ^b^	**<0.05**	0.70 ± 0.18 ^a^	1.03 ± 0.15 ^b^	0.53 ± 0.08 ^a^	**<0.05**	0.69 ± 0.12	0.55 ± 0.07	0.150
	Gamma globulin	0.5–1.2	0.64 ± 0.45 ^a^	1.30 ± 0.72 ^b^	0.77 ± 0.26 ^b^	**<0.05**	0.42 ± 0.16 ^a^	1.23 ± 0.35 ^b^	0.71 ± 0.39 ^a^	**<0.05**	0.65 ± 0.44	0.75 ± 0.21	0.269

* Transaminases AST: aspartate aminotransferase; transaminases ALT: alanine aminotransferase; and A/G ratio: albumin/globulin ratio.

## Data Availability

The datasets supporting the conclusions of this article are included within the article. Raw data are available from the corresponding author upon reasonable request.

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
