# Peer review of "Effect of Breed on Hematological and Biochemical Parameters of Apparently Healthy Dogs Infected with Zoonotic Pathogens Endemic to the Mediterranean Basin"

_animals, 2024, doi:10.3390/ani14111516_

Round 1

Reviewer 1 Report

Comments and Suggestions for Authors

The idea of this research is good, but many changes must be made in the manuscript, which shows sloppiness in writing. In addition, there are many grammar mistakes. The manuscript should be corrected for the English language.

L26: “L. infantum, R. rickesii“ full names. It is the first time they mentioned in the summary.

L33: “Metthods” is “Methods”

L33: “L. infantum, R. ricketsii, A. phagocytophilum, E. canis” full names. It is the first time they mentioned in the abstract.

L39: “CBC” full name. It is the first time they mentioned in the abstract.

L41: “MCH, A/G ratio” full names. It is the first time they mentioned in the abstract.

L61, 62, 63, 64: the names of the pathogens should be full-named. It is the first time they mentioned in the manuscript.

L74: “Sheperd” is “Shepherd”

L113-116: the pathogens should be written in italics

L116: What references did you use for the choice of the cut-offs? (titer ≥1:80 for Leishmania spp and titer≥1:50 for the others)

L118: “Hematological profile (complete blood count)” should be “Complete blood count (CBC)”

L120: “leukocytes concentration (WBC)” The correct is “white blood cells count (WBC)”

L121: “and white blood cells (WBC)” Why did you mention it again?

L123: “aspartate aminotransferase (GOT)” the correct is (AST). Please replace “GOT” with “AST” in whole manuscript

L128: “L. infantum” italics

L151-153: rewrite the paragraph. It does not make sense

L154-156: rewrite the sentence and include the number of the dogs, not only the percentages

L157-158: include the numbers of the dogs

Table 1: the total number of dogs that were affected is 44 and in the manuscript, you say that it is 42

L167: use the abbreviations, not the whole name. You have mentioned it before

L170: use the same abbreviation and name in the whole manuscript. Do not change it. GOT should be AST. GPT is ALT.

L177: “of white blood cells, platelets, and leukocytes concentration (WBC).” “White blood cells” is WBC. Why did you mention it again?

Table 2: “RBC: red blood cells concentration; MCV: mean corpuscular volume; MCH: mean corpuscular hemoglobin; MCHC: mean corpuscular hemoglobin concentration; WBC: leukocytes concentration.” as a footnote.

Table 3: “Different superscript in row indicates significant differences (p<0.05). Transaminases AST: aspartate aminotransferase; Transaminases ALT: alanine aminotransferase; A/G: albumin/globulin.” as a footnote.

Table 4: “Different superscript indicates significant differences (p<0.05). RBC: red blood cells concentration; MCV: mean corpuscular volume; MCH: mean corpuscular hemoglobin; MCHC: mean corpuscular hemoglobin concentration; WBC: leukocytes concentration. LI: L. infantum infection; AP: A. phagocytophilum infection; RR: R. ricketsii infection.” As a footnote

Table 4: at the column of CBC parameters, correct the units of measurements of WBC and the differential blood cells

Table 5: “Different superscript in row indicates significant differences 201

(p<0.05). Transaminases AST: aspartate aminotransferase; Transaminases ALT: alanine aminotrans-202

ferase; A/G: albumin/globulin. LI: L. infantum infection; AP: A. phagocytophilum infection; RR: R. ricketsii infection” as a footnote

L206: “component blood count (CBC)” is “complete blood cell” but you can write only the “CBC”

Figure 1: “L + R: L.infantum and R. ricketsii coinfection; L + A: L. infantum and A. phagocytophilum coinfection; R + A: R. ricketsii and A. phagocytophilum coinfection. A/G ratio: albumin/globulin ratio. Data show as mean ±” use the same abbreviations in the whole manuscript. Do not change them.

Figure 1: it would be better if you separated into 2 figures. The letters are too small that are not visible.

L230: only the abbreviation “CBC”

Figure 2: it would be better if you separated into 2 figures. The letters are too small that are not visible.

L244-247: pathogens in italics

L251-258: replace “GOT” with “AST”

Table 6: “Different superscript in row indicates significant differences (p<0.05). RBC: red blood cells concentration; MCV: mean corpuscular volume; MCH: mean corpuscular hemoglobin; MCHC: mean corpuscular hemoglobin concentration; WBC: leukocytes concentration.” as a footnote.

Table 7: “Different superscript in row indicates significant differences (p<0.05). Transaminases AST: aspartate aminotransferase; Transaminases ALT: alanine aminotransferase; A/G: albumin/globulin” as a footnote.

L330: “FCI” full name

L351: “and human infected by Plasmodium falciparum [62],” it is irrelevant

Comments on the Quality of English Language

there are many grammar mistakes 

Author Response

Thank you very much for your comments and suggestions. The answers have been added point by point in the attached file.

Reviewer 2 Report

Comments and Suggestions for Authors

Dear authors,

The article is interesting and provides additional data about the effect of breed on hematological and biochemical parameters of apparently healthy dogs for zoonotic pathogens in Sicily. Nonetheless, certain concerns must be resolved before the manuscript can move forward in the review process.

Greyhounds have unique hematologic and biochemical characteristics compared with non-Greyhound breeds. Total protein, albumin, globulin, creatinine, WBC, RBC, HGB, HCT, MCHC, etc. reference intervals differed from reference intervals of other breeds.

Therefore, in this study, the question arises whether changes in hematological and biochemical parameters in Cirneco dell’Etna with other breeds are due to the infection or due to the uniqueness of that breed. A comparison should be made with uninfected individuals of the same breed to draw a conclusion.

Another issue is the number of individuals of this breed out of 30 dogs, 15.8% were infected with mono and two infections, and 3.5% with three infections. Besides that, the number of dogs of non-Greyhound breeds is even smaller. Nowhere in the paper is it written how many dogs of other tested breeds are infected, considering that the initial number is 8 and 6, that number can only be lower and therefore the statistical analysis cannot be valid.

 Only if the number of dogs for this study is increased can it be considered for the next review.

I would suggest the authors to expand the study

-          - to a larger number of dogs of all breeds

-          - or to conduct research related only to Cirneco dell’Etna breed, with increased numbers. It could be enlightening to investigate the hematological and biochemical parameters in infected and non-infected groups, particularly given the breed's distinct and endemic nature. Also to make a comparison between the groups of this breed seropositive on L.infatum, R.ricketsii, and A.phagocytophylum. regarding hematological and biochemical parameters.

 Minor issue

Pay attention that species names are in italics (see introduction, materials and methods..).

Remove in abstract “Background”, “ Metthods”,  “ Results ”,   and “Conclusions”

Line 122: creatinine is mentioned 2 times, delete one.

 Line 162: please present the results (in tabular or graphical form) of the prevalence of the tested pathogens in all breeds.

Intervals in all tables - you must refer to another reference given that Cirneco dell’Etna has unique hematologic and biochemical characteristics compared with non-Greyhound breeds.

The authors detected only 2 individuals with three infections, therefore such a group cannot be included in the statistical analysis because a valid result cannot be obtained. For this reason, remove the column ”three infections” in table 2 and 3. For a similar reason, the results shown in tables 4 and 5 cannot be valid because there are 3 individuals in two groups (LI+AP; RR+AP).

Comments on the Quality of English Language

Only if the number of dogs for this study is increased can it be considered for the next review.

Author Response

(The authors gave the same response as above.)

Reviewer 3 Report

Comments and Suggestions for Authors

The manuscript “Effect of breed on hematological and biochemical parameters of
apparently healthy dogs for zoonotic pathogens endemics of the Mediterranean
basin” has addressed an important issue related to public health. It gives some good insights related to zoonosis in different breeds of dog with the explanations of various hematological and biochemical parameters.

I have few suggestions to improve the manuscript

1.       Line 81; Health dogs to be replaced by healthy dogs.

2.       Line 151-153: The sentences “Nineteen dogs were males (33.33%), and twenty-seven (47.37%) and twenty-one dogs (36.84.54%) were young and adult, respectively. All the animals were standard vaccination complete and only three (5.26%) were anti-Leishmania vaccination” does not make any sense. How many were females? If 27 were females then 19+27 give total of 56 animals however, 57 animals were included in the study. Please clarify.

3.       Line 206: Component Blood Count (CBC) should be replaced with Complete Blood Count (CBC)

4.       Line 249: Should mention the name of breed instead of this breed.

5.       Complete document should be checked for writing the scientific names in italic format

6.       Results should be written in more clear way. In my view a table mentioning breeds wit relation to type of pathogens found should be included.

7.       Given the dichotomous nature of the response variable and the study's goal of comparing different breeds, using logistic regression to calculate odds ratios is appropriate. However, the small sample sizes for some breeds could compromise the reliability of these comparisons. Increasing the sample sizes would help ensure more accurate and dependable results from the logistic regression analysis.

Best wishes.

Author Response

(The authors gave the same response as above.)

Round 2

Reviewer 2 Report

Comments and Suggestions for Authors

Thanks to the authors for the effort in rearranging the manuscript. However, issues are still not fixed after the first review.

As I wrote before, greyhounds possess unique hematologic and biochemical characteristics compared to non-greyhound breeds. Previous studies have reported that greyhounds have higher RBC,  HCT, HGB concentration, MCV, and MCHC when compared with values in non-greyhound dogs. Others have also reported lower plasma and serum protein concentrations and lower serum globulin concentrations. In this study, similar results were seen in infected Cirneco dell’Etna dogs, in comparison to other breeds.

The authors did not discuss how this breed can impact hematological and biochemical parameters in their discussion, which is an important factor to consider in interpreting the results.

Given that the number of infected Cirneco dell’Etna dogs is now clearly visible. I suggest that the authors exclude other breeds in the manuscript due to the scarcity of individuals and the distinctive characteristics of greyhounds. Analyze and compare infected Cirneco dell’Etna dogs with non-infected ones, and also compare infected dogs with their corresponding parasites.

Author Response

We have answered to the reviewer's suggestions point by point in the attached file.

Reviewer 3 Report

Comments and Suggestions for Authors

The manuscript entitled “Effect of breed on hematological and biochemical parameters of apparently healthy dogs infected for zoonotic pathogens endemics of the Mediterranean basin” has been much improved after revision. However, I have few suggestions as:

1.       Line: 156-157; Please replace “Nineteen dogs were males (33.33%) and thirty-eight females (66.67%) were included in this study” with “Nineteen male dogs (33.33%) and thirty-eight females (66.67%) were included in this study”

2.       Regarding statistical analysis by considering the limitations of the study you may not be able to use logistic regression in this case bootstrap method is a good alternative if you can perform that otherwise descriptive statistics is enough.

Best wishes.

Author Response

(The authors gave the same response as above.)
